# The Prognostic Role of True Radical Resection in Perihilar Cholangiocarcinoma after Improved Evaluation of Radial Margin Status

**DOI:** 10.3390/cancers14246126

**Published:** 2022-12-12

**Authors:** Mario De Bellis, Maria Gaia Mastrosimini, Simone Conci, Sara Pecori, Tommaso Campagnaro, Claudia Castelli, Paola Capelli, Aldo Scarpa, Alfredo Guglielmi, Andrea Ruzzenente

**Affiliations:** 1Department of Surgery, Dentistry, Gynecology and Pediatrics, Division of General and Hepatobiliary Surgery, University of Verona, G.B. Rossi University Hospital, P. le L.A. Scuro 10, 37134 Verona, Italy; 2Department of Diagnostics and Public Health, Section of Pathology, University of Verona, G.B. Rossi University Hospital, P. le L.A. Scuro 10, 37134 Verona, Italy

**Keywords:** surgical margin, radial margin, periductal margin, residual disease, hilar cholangiocarcinoma

## Abstract

**Simple Summary:**

A proper pathological examination of resected perihilar cholangiocarcinoma specimen should take into consideration both the ductal and the radial margin status. Unfortunately, current evidence shows that pathological reports offer a poor assessment of residual disease status, especially in Western centers. The ambiguity in reporting on surgical margins impedes correct staging, prognosis, and the consistent design of survival studies. The present study reviews the Verona (Italy) experience in surgical treatment of PHCC after improved evaluation of surgical margins status and consequently investigates the impact of true R0 (negative ductal and radial margin) on survival. Radial Margin positivity was the most frequent cause of R1, and multivariable analysis identifies residual disease status as the main independent factor affecting both RFS and OS. The improved evaluation of RM status could lead to a more accurate selection of patients for adjuvant therapy.

**Abstract:**

Background: The evaluation of surgical margins in resected perihilar cholangiocarcinoma (PHCC) remains a challenging issue. Both ductal (DM) and radial margin (RM) should be considered to define true radical resections (R0). Although DM status is routinely described in pathological reports, RM status is often overlooked. Therefore, the frequency of true R0 and its impact on survival might be biased. Objective: To improve the evaluation of RM status and investigate the impact of true R0 on survival. Methods: From 2014 to 2020, 90 patients underwent curative surgery for PHCC at Verona University Hospital, Verona, Italy. Both DM (proximal and distal biliary margin) and RM (hepatic, periductal, and vascular margin) status were evaluated by expert hepatobiliary pathologists. Patients with lymph-node metastases or positive surgical margins (R1) were candidates for adjuvant treatment. Clinicopathological and survival data were retrieved from an institutional database. Results: True R0 were 46% (41) and overall R1 were 54% (49). RM positivity resulted in being higher than DM positivity (48% versus 27%). Overall survival was better in patients with true R0 than in patients with R1 (median survival time: 53 vs. 28 months; *p* = 0.016). Likewise, the best recurrence-free survival was observed in R0 compared with R1 (median survival time: 32 vs. 15 months; *p* = 0.006). Multivariable analysis identified residual disease status as an independent prognostic factor of both OS (*p* = 0.009, HR = 2.68, 95% CI = 1.27–5.63) and RFS (*p* = 0.009, HR = 2.14, 95% CI = 1.20–3.83). Conclusion: Excellent survival was observed in true R0 patients. The improved evaluation of RM status is mandatory to properly stratify prognosis and select patients for adjuvant treatment.

## 1. Background

Perihilar cholangiocarcinoma (PHCC) is the most common type of biliary tract cancer and has a dismal prognosis with 5-year survival rates of 35–44% in high-volume centers [1,2,3] Surgery is the only treatment that can provide the chance for a cure. Surgical resection requires a major hepatectomy with en bloc resection of the caudate lobe and the extrahepatic bile duct, in addition to locoregional lymphadenectomy. Furthermore, concomitant vascular resections or pancreatoduodenectomy are performed to aim for a radical resection (R0). R0 is defined as the histological evidence of tumor-free margins and is a strong positive prognostic factor since it ensures long recurrence-free (RFS) and overall survival (OS) [4]. The correct evaluation of residual disease in resected PHCC must consider both the ductal (DM) and radial margin (RM) status [5]. DM status is determined by the proximal and distal biliary margins, whereas RM status is determined by the transection margin of the hepatic parenchyma along with the dissection margin of the hepatoduodenal ligament and the vascular margin. Unfortunately, recent studies [6,7] showed that pathological reports of resected PHCC offer a poor assessment of surgical margins, especially in Western centers where the completeness of pathology reports ranges from 10% to 45%. RM status is frequently overlooked, even though a positive RM is observed more often than a positive DM. Furthermore, the criteria for the definition of R0 are not univocal and the differences concern the length of tumor clearance [8]. For the above reason, the reported rates of R0 in published literature are very variable, ranging between 19% to 95% [9,10,11,12,13], and it is not always clear which surgical margins were evaluated and how their status was defined. Incomplete assessment of surgical margins status may overestimate R0 resections and thus prevent proper staging and comparison of survival studies. Our pathology service has gained extensive experience in the systematic evaluation of DM and RM status of resected PHCC by applying a standardized protocol for grossing and reporting. The aim of this study is to review our tertiary center experience in the surgical treatment of PHCC after improved evaluation of RM status and consequently investigate the impact of true R0 on survival.

## 2. Patients and Methods

### 2.1. Study Population

Consecutive patients who underwent curative intent resection for PHCC from 2014 through 2020 at the Division of General and Hepatobiliary Surgery, Verona University Hospital, Italy were identified from an institutional database. All surgical specimens were submitted to an improved pathological examination in order to properly identify and describe RM status. PHCC was defined as a biliary tumor involving the hepatic duct confluence according to the definition of the Japanese Society of Biliary Surgery (JSBS) [14]. Exclusion criteria were resection with macroscopic residual disease (R2), evidence of metastases including lymph-node metastases beyond the hepatoduodenal ligament, and excision of only the extrahepatic bile duct. Written informed consent was obtained from all patients before surgical procedure. Data collection and analysis were performed according to the institutional guidelines conforming to the ethical standards of the Helsinki Declaration, and the study was approved by local ethics committee.

### 2.2. Preoperative Management

The type of surgery was planned according to the hepatic location of the tumor, the presence of vascular invasion, the liver function, and the future remnant liver volume. Patients with extrahepatic disease or liver metastases were not candidates for surgery. Patients with tumor involvement of the portal vein and hepatic artery on the side of the future remnant liver without the possibility of a vascular reconstruction, extensive bilateral proximal infiltration beyond secondary biliary radicles, and/or massive extension into the liver parenchyma were deemed unresectable. MRI and CT scan with contrast enhancement were routinely performed for tumor staging. High-quality cross-sectional imaging provided essential information about vascular invasion and tumor location relative to the biliary tree. The longitudinal extent of ductal infiltration was also assessed by direct cholangiography (either endoscopic or transhepatic) or cholangioscopy with mapping biopsy and classified according to the Bismuth–Corlette classification. Selected patients also underwent a PET scan to evaluate the presence of extrahepatic disease. Jaundice patients underwent either endoscopic or transhepatic biliary drainage and surgery was performed after serum total bilirubin levels dropped to less than 3 mg/dL. Liver function was assessed by indocyanine green retention rate test at 15 min, and levels less than 14%/min were considered appropriate for major hepatectomy. When the future liver remnant volume was less than 35% of the total, portal vein embolization (PVE) was performed. Preoperative chemotherapy and/or radiotherapy was not routinely administrated.

### 2.3. Surgery

Resectability was assessed by abdominal exploration and intraoperative ultrasound. The parenchymal transection was performed using the ultrasonic surgical aspirator system with intermittent Pringle’s maneuver. Resection of liver segment I was always performed, whereas resection of the portal vein and/or hepatic artery was performed only when macroscopic vascular invasion was suspected. Combined pancreatoduodenectomy was performed in cases of tumor spreading towards the common bile duct or bulky node metastases around the pancreatoduodenal region. In order to easily evaluate the RM status and increase the chance of R0, we isolated the common bile duct towards the upper border of the pancreas and the hepatic artery as far as possible from the tumor, then, we proceeded towards the hilum, peeling the portal vein up to its confluence with complete en bloc excision of the bile duct and the fatty tissue of the hepatoduodenal ligament. Dissection of vessels was limited to the future remnant liver, avoiding unnecessary dissection and detachment of vessels from the peritumoral tissue if technically possible. Frozen sections of proximal and distal bile duct margins were performed in all cases. If positive DM, additional bile duct resection was performed as far as technically feasible to obtain R0. Lymphadenectomy was classified according to the classification of the JSBS [14]. Lymph nodes of the hepatoduodenal ligament (station 12), the proper hepatic artery (station 8), and the posterior surface of the head of the pancreas (station 13) were routinely retrieved. Interaortocaval lymph nodes (station 16) were retrieved only when macroscopically abnormal.

### 2.4. Pathological Evaluation

The specimens were fixed in 4% buffered formalin for approximately 24–48 h. Then, all surgical margins were stained according to a color code for easy recognition under the microscope. Briefly, the resection margin of the distal bile duct, proximal bile duct(s), hepatic artery, and portal vein were sampled. After, the specimens were sliced in 3- to 5-mm-thick slices following an axial plane perpendicular to the extrahepatic bile duct axis up to the biliary confluence in order to leave the periductal tissue surrounding the tumor that is the dissection margin of the hepatoduodenal ligament (retroperitoneal surface) intact. Crossing the biliary confluence, the slicing carried on in a coronal plane to better appreciate the tumor growth along the intrahepatic bile ducts and identify suspected infiltration of the hepatic parenchymal transection margin. The samples were embedded in paraffin and prepared for microscopic examination using hematoxylin and eosin staining. Pathological reports were drafted according to the International Collaboration on Cancer Reporting [15] and the TNM Classification of Malignant Tumors by the International Union Against Cancer (8th edition, 2016). DM was classified as positive when invasive carcinoma was identified at the proximal or distal bile duct margin, literally at the edge of surgical cut. DM with different grade of dysplasia up to carcinoma in situ was considered as negative [16]. RM was classified as positive when tumor cells were identified less than 1 mm from the transection plane of the hepatic parenchyma, the dissection plane of the hepatoduodenal ligament, or the vascular stumps. The involvement of the peritoneal surface of the hepatoduodenal ligament by the tumor was not considered as a margin since the surgeon does not cut or dissect any tissue [15]. Finally, surgery was defined as (true) R0 if both DM and RM status was negative or as R1 if either DM or RM was positive.

### 2.5. Follow-Up

The decision to administer adjuvant therapy was made by a multidisciplinary team. In principle, patients with lymph node metastases or R1 resection are candidates for chemotherapy and/or radiotherapy. All patients underwent surveillance for recurrence with CT scan or MRI usually every 4–6 months. In cases of questionable radiological diagnosis, a PET scan was performed. Pathologic confirmation was not routinely carried out. When feasible, recurrence was treated with either surgery, chemotherapy, or radiotherapy.

### 2.6. Statistical Analysis

Statistical analysis was performed with SPSS (version 20.0, SPSS Inc.) (Amok, NY, USA). Categorical variables were expressed in numbers and percentages and were compared among groups using Fisher’s exact test or Pearson’s chi-square test, as appropriate. Continuous variables were expressed as median values with the interquartile range (IQR) and were compared using Mann–Whitney U test. RFS and OS curves were constructed using the Kaplan–Meier method from the time of surgery to the time of recurrence, death, or last follow up. Differences between survival probabilities were compared using the log-rank test excluding from the analysis postoperative deaths, defined as any deaths occurring within 90 days of surgery or during the same hospital stay, whenever it occurred. A multivariate analysis was performed using the Cox proportional hazards model to identify prognostic factors by backward elimination. *p* ≤ 0.05 was considered to indicate statistical significance. 

## 3. Results

### 3.1. Baseline Characteristics of Our Cohort

During the study period, 100 consecutive patients with PHCC underwent surgery. Of these, six (6%) patients were excluded due to intraoperative detection of either liver or peritoneal metastases, three (3%) due to lymph-node metastases beyond the hepatoduodenal ligament, and one (1%) due to R2 resection. The remaining 90 patients were included in this study. Median age was 70 (IQR, 62–76) years, and 60 (67%) patients were male. Median CEA and CA19-9 were 2 (IQR, 2–2.8) ng/mL and 222.8 (IQR, 47.5–918.8) U/mL, respectively. Overall, 77 (86%) patients underwent preoperative biliary drainage, namely 54 (60%) percutaneous transhepatic and 23 endoscopic biliary drainage (26%). Radiologic imaging showed biliary strictures as Bismuth type IV in 32 (36%) patients. Moreover, 13 (14%) patients received portal vein embolization; 49 (54%) patients underwent left hepatectomy; 28 (31%) right hepatectomy; 6 (7%) left trisectionectomy; 4 (4%) mesohepatectomy; 2 (2%) right trisectionectomy; and 1 (1%) hepatopancreatoduodenectomy. Vascular resection was performed in 15 (17%) patients, namely 14 (16%) portal vein resection and 1 (1%) hepatic artery resection. 

### 3.2. Histopathological Findings

Intraoperative frozen section analysis of proximal and distal ductal margin demonstrated invasive carcinoma in 30 (33%) patients. In one (1%) case, additional resection was not technically feasible. Of 29 (32%) patients who had additional bile duct resection, only 6 (7%) achieved R0 on final histological examination. Median tumor diameter was 2.5 (IQR, 1.7–3.5) cm, and 46 (51%) patients had pT3/4 tumor. Poorly differentiated or undifferentiated adenocarcinoma was noticed in 25 (28%) patients. Fifty-one (57%) patients had positive lymph nodes and median number of lymph node harvested was 9 (IQR, 6–13). Two (2%) patients had PHCC stage 1; 21 (23%) stage II; 11 (12%) stage IIIA; 5 (6%) stage IIIB; 41 (46%) stage IIIC; and 10 (11%) stage IVA. Demographic, clinicopathological, and operative features were summarized in Table 1. 

### 3.3. Surgical Margins

A total of 41 (46%) patients underwent R0, whereas the remaining 49 (54%) patients had R1. In particular, DM positivity was observed in 24 (27%) and RM positivity in 43 (48%) patients. The site of positive DM was proximal bile duct in 24 (27%) patients and distal bile duct in 5 (6%) patients. The site of positive RM was the periductal tissue in 39 (43%) patients, the liver parenchyma in 11 (12%) patients, and the vascular stumps in 4 (4%) patients (Figure 1). Six (7%) patients had positive DM alone, 18 (20%) had both positive DM and positive RM, and 25 (28%) had positive RM alone. 

### 3.4. Short- and Long-Term Outcomes

Morbidity was 71% (64), and major complications (Dindo grade≥ 3) were observed in 32% (29) of cases. The main complications were biliary fistula and post-hepatectomy liver failure, which occurred in 22 (24%) and 22 (24%) patients, respectively. 90-day mortality was 6% (5). Fifty-five (61%) patients underwent adjuvant therapy. Different regimens were used for chemotherapy, and the median number of treatment cycles was 6 (IQR, 2-8). Regarding radiotherapy, the radiation dose of 45Gy was administered in 25 fractions. In our cohort, the rate of OS at 1-, 3-, and 5-year was 88%, 60%, and 36%, respectively; the rate of RFS at 1-, 3-, and 5-year was 64%, 28, and 13%, respectively. The median follow-up time was 41 months. 

Overall survival was better in patients with R0 than in patients with R1 (median survival time (MST) 53 vs. 28 months; *p* = 0.016; hazard ratio (HR) 2.42, 95% confidence interval (CI) 1.18–4.95). Likewise, the best RFS was observed in R0 compared with R1 (MST 32 vs. 15 months; *p* = 0.006; HR 2.23, 95% CI 1.26–3.94). OS and RFS curves are shown in Figure 2. 

RM status, rather than DM status, was associated with OS (*p* = 0.013) and RFS (*p* = 0.031). Patients with negative RM compared to those with positive RM showed both prolonged OS (MST 53 vs. 28 months; *p* < 0.017; HR 2.35, 95% CI 1.17–4.75) and RFS (MST 32 vs. 15 months; *p* < 0.037; HR 1.80, 95% CI 1.04–3.12) (Figure 3).

Survival probabilities were also analyzed according to the operative procedures performed (extended versus non-extended hepatectomies, left-sided versus right-sided hepatectomies), but no significant differences were observed. Conversely excellent survivals were observed in patients with both negative lymph-nodes and surgical margins (Figure 4).

Residual disease status was an independent prognostic factors for both OS (*p* = 0.009, HR = 2.68, 95% CI = 1.27–5.63) and RFS (*p* = 0.009, HR = 2.14, 95% CI = 1.20–3.83). Multivariable analyses are reported in Table 2 and Table 3.

## 4. Discussion

The improved evaluation of RM status in resected PHCC allows to detect true RO. The present study shows that RM positivity is the most frequent cause of R1, and multivariable analysis identifies residual disease status as the main independent factor affecting both RFS and OS.

Unfortunately, the pathological reports of several Western centers do not provide a thorough assessment of all surgical margins in resected PHCC. A French multi-institutional survey [6] found that RM status was frequently overlooked; indeed, periductal soft tissue circumferential margin, vascular margin, and liver margin were assessed in only 10%, 13%, and 20% of cases, respectively. Likewise, a Dutch audit [7] demonstrated that residual disease status was unclear in 29% of cases and could be re-classified from R0 to R1 in 15%.

Our hepatobiliary pathologists have gained more than 10 years of experience with the systematic evaluation of both DM and RM status developing a standardized protocol for grossing and reporting according to the ICCR guidelines [15]. We believe that the correct evaluation of RM status is hindered by the complexity of PHCC specimen and above all by the lack of familiarity with the identification of the periductal circumferential margin in the soft tissue of the hepatoduodenal ligament. In fact, the periductal margin is a dissection plane, unlike the ductal, hepatic, and vascular margins, which are resection margins.

Stremitzer et al. [17] sought to investigate the prognostic role of the isolated positive periductal dissection margin by retrospectively reviewing the data of 83 patients from two European Centers over a period of 10 year (2006–2016). The authors considered DM and hepatic transection margin as the “surgical margin” and the interface of the extrahepatic bile duct with the surrounding lymphatic/fatty tissue as the “circumferential margin”. The median OS in patient with R0, isolated positive circumferential margin, and positive surgical margins was 45.6, 32.7, and 14.5 months, respectively (*p* = 0.011), whereas the median RFS showed no statistically significant differences. Both positive isolated circumferential and positive surgical margins were predictors of poor OS according to the multivariable Cox regression analysis. 

Mueller et al. identify R0 ≥ 56.7% and R1 ≤ 43% as benchmark cut-offs for PHCC surgery, however, it is not clear whether RM status was assessed in all the high-volume centers that participated in the study. In our cohort, if the assessment of surgical margins was incomplete, in other words by neglecting the condition of RM status, the rough R0 survival curve became steeper than the true R0 (median survival time 40 vs. 53 months). Therefore, 25 (28%) cases could be misclassified as R0 (Figure 5).

Very few studies have specifically addressed the issue of RM status in resected PHCC [8,18,19], and the group of Nagoya University, Japan, published the largest series so far [5]. This tertiary level Eastern Center retrospectively analyzed 478 consecutive cases over a period of 5 years (2001–2006) and reported 18% (85) of R1. RM positivity was 11% (52) and resulted in the most common cause of residual disease. In particular, periductal margin positivity was 4% (20), hepatic margin 4% (20), and vascular 3% (12). The reason for these surprisingly low percentages could be explained by the use of a different surgical strategy or patient selection. Shinohara et al. [5] found that the survival time of patients with positive RM was significantly shorter compared to that of R0 resection (median survival time 2.1 vs. 4.9 years; *p* < 0.001; HR = 2.06, 95% CI = 1.49–2.84). Instead, no survival difference was noted between RM and DM positivity. Both RM (HR = 1.48; 95% CI = 1.05–2.08; *p* = 0.023) and DM positivity were independent prognostic factors of poor OS. 

D’Amico et al. [20] retrospectively analyzed 75 patients over a period of 12 years (2005–2017) and reported 45% (34) of R1. The authors confirmed a higher rate of RM positivity (35%, *n* = 26) compared to DM positivity (23%, *n* = 17) and observed, instead, that patients with isolated positive RM (23%, *n* = 17) had statistically similar OS and RFS to patients with R0 (55%, *n* = 41). The latter study, though, had a smaller sample of only 9% (7) of pT3/4 tumors, and included six PHCC Bismuth type I who did not undergo hepatic resection.

Recent systematic reviews and meta-analyses of resected PHCC reported that positive surgical margins are prognostic factors of poor survival [4,21], however, even among resected patients with declared negative surgical margins, short RFS and OS are frequently observed [9,11]. We speculate that this finding may be due to the lack of a comprehensive evaluation of RM status. We also hypothesized that short RM tumor clearance might be related to a high rate of local recurrence and short survivals. If we select a cut-off of 0 mm tumor clearance to define RM positivity, we still observe significantly different survivals curves by RM and residual disease status. Instead, selecting a cut-off of less than 2 mm, we observe no survival differences. Therefore, although the cut-off of 1 mm tumor clearance arbitrarily used to define RM positivity seems to be appropriate, the authors are aware that no definitive conclusion can be drawn by these findings since the sub-cohorts with different RM tumor clearance are small (Table 4).

Recently, D’Souza et al. [22] pointed out that there is no universal agreement on the definition of what constitutes R0 and how wide the tumor-free margin should be, but failed to prove a significant impact of the chosen definition of R0 (>0 mm or >1 mm to cancer-involved resection margin or dissection plane) on OS or RFS. Seyama et al. [23], in a series of 58 consecutive major hepatectomies, found that a surgical margin over 5 mm provided a significantly better survival. On the other hand, the survival rate after R0 with a narrow margin (<5 mm) was nearly the same as after R1.

The main limitations of the present study are the small sample, the analysis of data from a single center, and the use of different regimens of chemotherapy. Nevertheless, our findings are of great value, since the evaluation of RM status was assessed prospectively and the tumor distance from surgical margin was precisely classified. Thus, the risk of underestimating RM positivity is low and the frequency of reported true R0 is reliable. Lastly, the present study reported the largest Western data so far and is the first to demonstrate RM positivity as a prognostic factor of both poor OS and RFS. The improved evaluation of RM status could lead to a more accurate selection of patients for adjuvant therapy. In fact, little evidence exists about survival benefit of chemo/radiotherapy in patients who underwent PHCC resection with R1 [24,25], and the lack of information about RM status jeopardizes the credibility of survival studies.

## 5. Conclusions

Evidence of negative surgical margins is a strong predictor of good survival, hence both the RM and DM status need to be analyzed in PHCC specimens. Only by properly distinguishing patients between true R0 and R1, the prognosis can be adequately stratified. Furthermore, since residual disease status is one of the main criteria for the administration of adjuvant treatments, the improved evaluation of RM status is mandatory to compare survival studies.

## Figures and Tables

**Figure 1 cancers-14-06126-f001:**
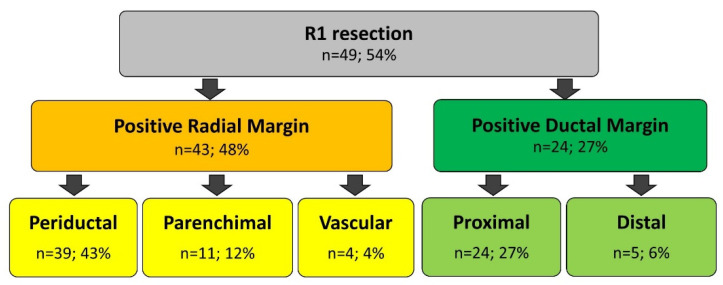
Comprehensive analysis of surgical margins positivity in R1 resection.

**Figure 2 cancers-14-06126-f002:**
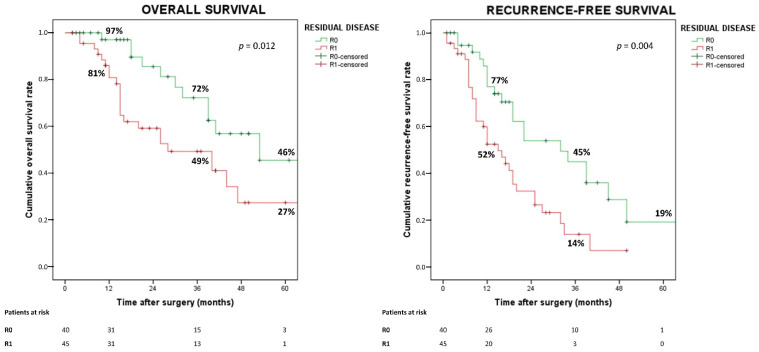
Overall and recurrence-free survival according to residual disease status.

**Figure 3 cancers-14-06126-f003:**
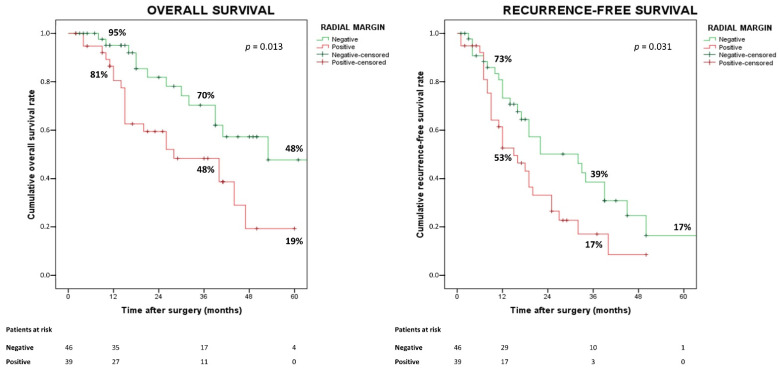
Overall and recurrence-free survival according to radial margin status.

**Figure 4 cancers-14-06126-f004:**
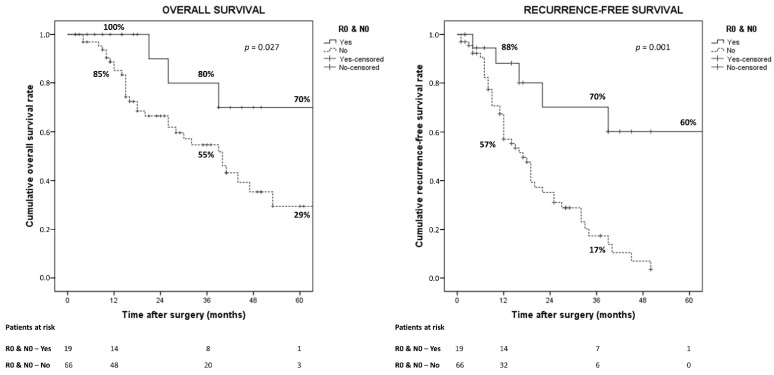
Overall and recurrence-free survival according to residual disease and lymph-nodes status.

**Figure 5 cancers-14-06126-f005:**
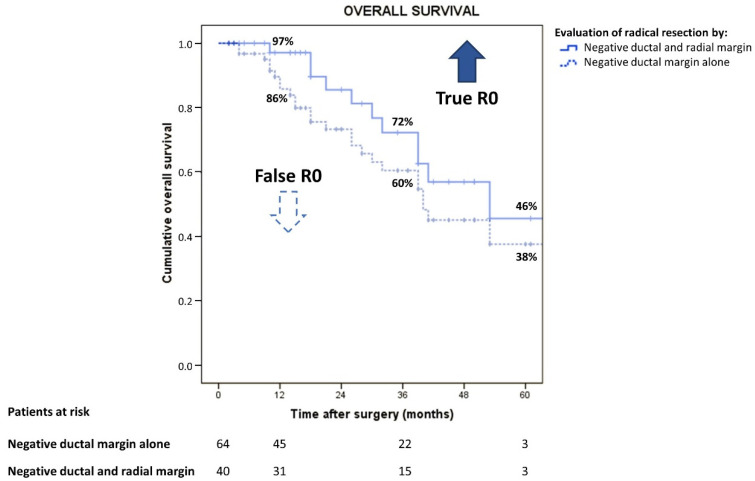
Overall survival in perihilar cholangiocarcinoma by different evaluation of radical resection.

**Table 1 cancers-14-06126-t001:** Demographic, clinicopathological, and operative features of 90 perihilar cholangiocarcinoma patients compared according to residual disease and radial margin status.

Variables	R0	R1	*p* Value	RM−	RM+	*p* Value
	(*n* = 41)	(*n* = 49)		(*n* = 47)	(*n* = 43)	
Age, years	69 (58–77)	70 (65–76)	0.279	70 (62–76)	70 (63–76)	0.475
Gender, male	22 (54)	38 (78)	0.017	27 (51)	33 (77)	0.052
CEA ng/mL	2 (2–2.7)	2 (1.6–3.4)	0.967	2 (2–2.7)	2 (1.7–4)	0.481
CA19–9 U/mL	81 (26–538)	429 (93–1194)	0.012	75 (27–587)	439 (156–1151)	0.014
Preoperative biliary drainage						
transhepatic	25 (61)	29 (59)	0.863	28 (60)	26 (60)	0.931
endoscopic	8 (20)	13 (27)	0.433	11 (23)	10 (23)	0987
Bismuth classification			0.069			0.093
Type II	4 (10)	0 (0)		4 (9)	0 (0)	
Type IIIa	6 (15)	13 (27)		9 (19)	10 (23)	
Type IIIb	18 (44)	17 (35)		21 (45)	14 (33)	
Type IV	13 (32)	19 (39)		13 (28)	19 (44)	
PVE	5 (12)	8 (16)		7 (15)	6 (14)	
Type of resection			0.285			0.349
Left-sides hepatectomy	29 (68)	27 (55)		32 (68)	24 (56)	
Right-sides hepatectomy	11 (29)	19 (39)		14 (30)	16 (37)	
Mesohepatectomy	1 (2)	3 (6)		1 (2)	3 (7)	
Vascular resection	4 (10)	11 (22)	0.108	5 (11)	10 (23)	0.109
Histopathological tumor grade			0.047			0.153
Well/moderately	33 (80)	30 (61)		36 (77)	27 (63)	
Poorly/undifferentiated	8 (20)	19 (39)		11 (23)	16 (37)	
Satellitosis	1 (2)	8 (16)	0.029	1 (2)	8 (19)	0.009
Tumor diameter, cm	2 (1.5–3)	3 (2–4)	0.025	2 (1.5–3.5)	3 (2–4)	0.044
AJCC pT classification			0.094			0.202
T1/T2	24 (59)	20 (41)		26 (55)	18 (42)	
T3/T4	17 (41)	29 (59)		21 (45)	25 (58)	
Perineural invasion	39 (95)	45 (92)	0.534	43 (91)	41 (95)	0.463
Ductal margin positivity	0 (0)	24 (49)	<0.0001	6 (13)	18 (42)	0.002
Lymph node metastasis	21 (51)	30 (61)	0.340	25 (53)	26 (60)	0.487
Lymph node harvasted	8 (6–13)	10 (7–16)	0.074	8 (5–13)	10 (7–16)	0.082
Major complication (Dindo ≥ 3)	9 (22)	20 (41)	0.129	11 (23)	18 (42)	0.128
90-day/in-hospital Mortality	1 (2)	4 (8)	0.238	1 (2)	4 (9)	0.138
Adjuvant chemo/radiotherapy	23 (56)	32 (65)	0.372	29 (62)	26 (60)	0.904

Categorical variables are expressed in numbers and percentages. Continuous data are expressed as median values and interquartile range. Abbreviations: BMI, Body Mass Index; ASA, American Society of Anesthesiologists; CEA, Carcino-Embryonic Antigen; CA19-9, Carbohydrate Antigen 19-9; PVE, Portal Vein Embolization; PI, periductal Infiltration; IG, Intraductal growth, MF, Mass Forming; and AJCC, American Joint Committee on Cancer staging system 8th edition.

**Table 2 cancers-14-06126-t002:** Univariable and multivariable analysis of overall survival.

Variables	OS (%)				Univariable	Multivariable	
	*n*	1-Year	3-Year	5-Year	*p* Value	HR (95% CI)	*p* Value
Gender					0.049		
Female	29	96	74	54			
Male	56	89	53	28			
Bismuth type IV					0.875		
No	53	94	60	40			
Yes	32	79	58	21			
Side of hepatectomy *					0.489		
Left	54	90	61	37			
Right	28	87	54	22			
Combined vascular resection					0.211		
No	73	89	63	38			
Yes	12	80	41	21			
Tumor diameter, cm					0.213		
<3	45	90	65	39			
≥3	40	86	54	33			
Histopathological tumor grade					0.042		0.045
Well/moderately	60	95	68	35		1	
Poorly/undifferentiated	25	72	37	37		2.07 (1.02–4.21)	
Perineural invasion					0.405		
No	6	63	31	/			
Yes	79	90	60	37			
AJCC pT classification					0.086		
T1/T2	42	92	71	59			
T3/T4	43	84	50	22			
Residual disease status					0.012		0.009
R0	40	97	72	46		1	
R1	45	81	49	27		2.68 (1.27–5.63)	
Lymph node					0.015		0.008
N0	36	93	73	63		1	
N+	49	84	49	20		2.73 (1.29–5.75)	
Dindo classification ≥ 3					0.960		
No	56	91	61	33			
Yes	29	82	55	/			
Adjuvant therapy					0.823		
No	30	85	63	38			
Yes	55	90	58	35			

Five postoperative deaths, defined as any deaths occurring within 90 days of surgery or during the same hospital stay, whenever it occurred, were excluded from the analysis. * Three mesohepatectomy were excluded. Abbreviations: AJCC, American Joint Committee on Cancer staging system 8th edition.

**Table 3 cancers-14-06126-t003:** Univariable and multivariable analysis of recurrence-free survival.

Variables	RFS (%)				Univariable	Multivariable	
	*n*	1-Year	3-Year	5-Year	*p* Value	HR (95% CI)	*p* Value
Gender					0.976		
Female	29	62	30	/			
Male	56	64	27	9			
Bismuth type IV					0.583		
No	53	62	23	/			
Yes	32	66	35	14			
Side of hepatectomy *					0.548		
Left	54	60	29	/			
Right	28	75	20	/			
Combined vascular resection					0.122		
No	73	64	33	15			
Yes	12	62	/	/			
Tumor diameter, cm					0.064		
<3	45	70	38	14			
≥3	40	56	19	/			
Histopathological tumor grade					0.227		
Well/moderately	60	69	26	12			
Poorly/undifferentiated	25	51	33	/			
Perineural invasion					0.932		
No	6	50	25	/			
Yes	79	65	29	12			
AJCC pT classification					0.014		0.030
T1/T2	42	67	48	/		1	
T3/T4	43	61	11	4		1.85 (1.05–3.24)	
Residual disease status					0.004		0.009
R0	40	77	45	19		1	
R1	45	52	14	/		2.14 (1.20–3.83)	
Lymph node					0.069		
N0	36	67	38	33			
N+	49	61	20	/			
Dindo classification ≥ 3					0.884		
No	56	66	27	13			
Yes	29	59	38	/			
Adjuvant therapy					0.076		
No	30	63	38	31			
Yes	55	64	22	/			

Five postoperative deaths, defined as any deaths occurring within 90 days of surgery or during the same hospital stay, whenever it occurred, were excluded from the analysis. * Three mesohepatectomy were excluded. Abbreviations: AJCC, American Joint Committee on Cancer staging system 8th edition.

**Table 4 cancers-14-06126-t004:** Definition of positive radial margin according to different tumor clearances and reclassification of residual disease status with the corresponding 5-year survivals rates.

RM Clearance	RM−	RM+	5-Year OS RM−/RM+	*p* Value	5-Year RFS RM−/RM+	*p* Value	R0	R1	5-Year OS R0/R1	*p* Value	5-Year RFS R0/R1	*p* Value
0 mm	68 (76%)	22 (24%)	39/26%	0.012	14/0%	0.053	50 (56%)	40 (44%)	25/42%	0.015	20/0%	0.003
<1 mm	47 (52%)	43 (48%)	48/19%	0.013	17/0%	0.031	40 (44%)	50 (56%)	28/46%	0.014	19/0%	0.006
<2 mm	8 (9%)	82 (91%)	50/32%	0.279	17/0%	0.393	8 (9%)	82 (91%)	32/50%	0.279	17/0%	0.393

## Data Availability

The data presented in this study are available on request from the corresponding author. The data are not publicly available due to privacy restrictions according to Italian law.

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
