# Peer review of "The Prognostic Role of True Radical Resection in Perihilar Cholangiocarcinoma after Improved Evaluation of Radial Margin Status"

_cancers, 2022, doi:10.3390/cancers14246126_

Round 1
Reviewer 1 Report
I read with interest the study by De Bellis and Collagues. The study is well conducted and highlighted interesting topics on the prognostic role of radical resection in perihilar cholangiocarcinoma.
I have some concerns and some minor revisions are needed:
- please indicate how were the patients studied before surgery and how was the assessment of PHCC done
- how were the patients drained before surgery? were the route of drainage considered as predictor both for R0 resection and RM or DM positivity
- over the years the use of peroral cholangioscopy showed a possible role for the evaluation of the extension of cholangiocarcinoma. Moreover, in the last two years a short version of Cholangioscopes, designed for the percutaneous and surgical approach has been introducted of the market. Please discuss if there could be a role of pre-operative or intra-operative cholangioscopy for a better assessment of neoplastic extension in order to reduce positive ductal margin.
Author Response
I read with interest the study by De Bellis and Collagues. The study is well conducted and highlighted interesting topics on the prognostic role of radical resection in perihilar cholangiocarcinoma.
The authors thank you for your comment.
I have some concerns and some minor revisions are needed:
- please indicate how were the patients studied before surgery and how was the assessment of PHCC done
Dear Reviewer, we edited “patients and method section” (2.2 preoperative management) according to your suggestions.
The type of surgery was planned according to the hepatic location of the tumor, the presence of vascular invasion, the liver function, and the future remnant liver volume. Patients with extrahepatic disease or liver metastases were not candidates for surgery. Patients with tumor involvement of the portal vein and hepatic artery on the side of the future remnant liver without the possibility of a vascular reconstruction, extensive bilateral proximal infiltration beyond secondary biliary radicles and/or massive extension into the liver parenchyma were deemed unresectable. MRI and CT scan with contrast enhancement were routinely performed for tumor staging. High-quality cross-sectional imaging provided essential information about vascular invasion and tumor location relative to the biliary tree. The longitudinal extent of ductal infiltration was also assessed by direct cholangiography (either endoscopic or transhepatic) or cholangioscopy with mapping biopsy and classified according to the Bismuth-Corlette classification. Selected patients also underwent PET scan to evaluate presence of extrahepatic disease. Jaundice patients underwent either endoscopic or transhepatic biliary drainage and surgery was performed after serum total bilirubin levels dropped to less than 3 mg/dL. Liver function was assessed by indocyanine green retention rate test at 15 min and levels less than 14 %/min were considered appropriate for major hepatectomy. When the future liver remnant volume was less than 35% of the total, portal vein embolization (PVE) was performed. Preoperative chemotherapy and/or radiotherapy was not routinely administrated.
- how were the patients drained before surgery? were the route of drainage considered as predictor both for R0 resection and RM or DM positivity
Thank you for your question.
Jaundice patients underwent either endoscopic or transhepatic biliary drainage and surgery was performed after serum total bilirubin levels dropped to less than 3 mg/dL”. We edited the “results section” (3.1. Baseline characteristics of our cohort) as follow: Seventy-seven (86%) patients underwent biliary drainage, namely 54 (60%) percutaneous transhepatic and 23 endoscopic biliary drainage (26%). We also added in table 1. (Demographic, clinicopathological, and operative features of 90 perihilar cholangiocarcinoma patients compared according to residual disease and radial margin status) the variables “transhepatic” and “endoscopic” preoperative biliary drainage. In our cohort, the route of drainage was not a predictor of positive surgical margins.
- over the years the use of peroral cholangioscopy showed a possible role for the evaluation of the extension of cholangiocarcinoma. Moreover, in the last two years a short version of Cholangioscopes, designed for the percutaneous and surgical approach has been introducted of the market. Please discuss if there could be a role of pre-operative or intra-operative cholangioscopy for a better assessment of neoplastic extension in order to reduce positive ductal margin.
The authors believe that peroral cholangioscopy has a role in the assessment of longitudinal tumor growth and therefore provide valuable information for surgical planning, especially in PHC Bismuth type IV. Currently we use preoperative peroral cholangioscopy with mapping biopsy, but we have little experience with intra-operative cholangioscopy in PHC. Although the authors agree with the reviewer that the use of cholangioscopy may reduce ductal margin positivity, the authors do not have sufficient data to discuss.
Reviewer 2 Report
Figure 4 is the most important in this paper, but are the survival curve lines reversed? Please check.
Author Response
Dear Reviewer,
thank you for yuor advice.
We checked figure 4 and we think that it is fine. Patients with both negative lymph-nodes (N0) and negative surgical margins (R0) have better survivals curves.
Reviewer 3 Report
In the manuscript entitled “The prognostic role of true radical resection in perihilar cholangiocarcinoma after improved evaluation of radial margin status” de Bellis et al. report a study to improve the assessment of radial margin resection status in resected perihilar cholangiocarcinoma and to investigate the effect of true R0 on survival.
This is the first assessment of its kind that prospectively assesses radial margin status with precise classification of tumor distance from the surgical margin. Although the results are quite interesting, the manuscript has some shortcomings that preclude an adequate and complete interpretation of these data. These shortcomings significantly affect the results and the authors' assessment of the aim of the study.
A serious problem of this prospective work is the incomplete information provided in the methods part on patient recruitment and resectability assessment. How much did the investigator's selection bias influence the decision to perform resection in advanced cases? Please provide further information on the patient characteristics of the patients examined compared to the entire cohort. This is particularly troublesome as this is one of the main parts of the study. More details should be given in this section to improve understanding. Authors should revise this section accordingly. The present study reports on the most extensive western data to date and is the first to show radial margin positivity as a prognostic factor for both poor overall survival and recurrence-free survival. How many patients had frozen sections of the proximal and distal bile duct margins performed? Please indicate in how many cases was an additional bile duct resection necessary and in how many cases was this not technically feasible? Please specify. Throughout the whole text there are many spelling and grammatical errors that disturb the article‘s informative value, e.g. parenchimal instead of parenchymal in figure 1. Accordingly, several statements need revision by a native speaker.
Author Response
In the manuscript entitled “The prognostic role of true radical resection in perihilar cholangiocarcinoma after improved evaluation of radial margin status” de Bellis et al. report a study to improve the assessment of radial margin resection status in resected perihilar cholangiocarcinoma and to investigate the effect of true R0 on survival.
This is the first assessment of its kind that prospectively assesses radial margin status with precise classification of tumor distance from the surgical margin. Although the results are quite interesting, the manuscript has some shortcomings that preclude an adequate and complete interpretation of these data. These shortcomings significantly affect the results and the authors' assessment of the aim of the study.
The authors thank you for your comments
A serious problem of this prospective work is the incomplete information provided in the methods part on patient recruitment and resectability assessment. How much did the investigator's selection bias influence the decision to perform resection in advanced cases? Please provide further information on the patient characteristics of the patients examined compared to the entire cohort. This is particularly troublesome as this is one of the main parts of the study. More details should be given in this section to improve understanding. Authors should revise this section accordingly.
The authors thank you for your advice and revised “patients and methods” section providing further information about patient selection and resectability assessment.
Study population: Consecutive patients who underwent curative intent resection for PHCC from 2014 through 2020 at the Division of General and Hepatobiliary Surgery, Verona University Hospital, Italy were identified from an institutional database… Exclusion criteria were resection with macroscopic residual disease (R2), evidence of metastases including lymph-node metastases beyond the hepatoduodenal ligament, and excision of only the extrahepatic bile duct…
Preoperative management: The type of surgery was planned according to the hepatic location of the tumor, the presence of vascular invasion, the liver function, and the future remnant liver volume. Patients with extrahepatic disease or liver metastases were not candidates for surgery. Patients with tumor involvement of the portal vein and hepatic artery on the side of the future remnant liver without the possibility of a vascular reconstruction, extensive bilateral proximal infiltration beyond secondary biliary radicles and/or massive extension into the liver parenchyma were deemed unresectable. MRI and CT scan with contrast enhancement were routinely performed for tumor staging. High-quality cross-sectional imaging provided essential information about vascular invasion and tumor location relative to the biliary tree. The longitudinal extent of ductal infiltration was also assessed by direct cholangiography (either endoscopic or transhepatic) or cholangioscopy with mapping biopsy and classified according to the Bismuth-Corlette classification. Selected patients also underwent PET scan to evaluate presence of extrahepatic disease. Jaundice patients underwent either endoscopic or transhepatic biliary drainage and surgery was performed after serum total bilirubin levels dropped to less than 3 mg/dL. Liver function was assessed by indocyanine green retention rate test at 15 min and levels less than 14 %/min were considered appropriate for major hepatectomy. When the future liver remnant volume was less than 35% of the total, portal vein embolization (PVE) was performed. Preoperative chemotherapy and/or radiotherapy was not routinely administrated.
The authors also added in “results” section the following sentence.
During the study period, 100 consecutive patients with PHCC underwent surgery. Of these, 6 (6%) patients were excluded due to intraoperative detection of either liver or peritoneal metastases, 3 (3%) due to lymph-node metastases beyond the hepatoduodenal ligament, and 1 (1%) due to R2 resection. The remaining 90 patients were included in this study.
The present study reports on the most extensive western data to date and is the first to show radial margin positivity as a prognostic factor for both poor overall survival and recurrence-free survival. How many patients had frozen sections of the proximal and distal bile duct margins performed? Please indicate in how many cases was an additional bile duct resection necessary and in how many cases was this not technically feasible? Please specify.
Thank you for your questions.
We edited the manuscript according to your suggestions as follow.
Patients and Methods: “Frozen sections of proximal and distal bile duct margins were performed in all cases. If positive DM, additional bile duct resection was performed as far as technically feasible to obtain R0”.
Results: “Intraoperative frozen section analysis of proximal and distal ductal margin demonstrated invasive carcinoma in 30 (33%) patients. In 1 (1%) case, additional resection was not technically feasible. Of 29 (32%) patients who had additional bile duct resection, only 6 (7%) achieved R0 on final histological examination”.
Throughout the whole text there are many spelling and grammatical errors that disturb the article‘s informative value, e.g. parenchimal instead of parenchymal in figure 1. Accordingly, several statements need revision by a native speaker.
We reviewed the text and corrected the spelling and grammatical errors.
Round 2
Reviewer 3 Report
In the manuscript entitled “The prognostic role of true radical resection in perihilar cholangiocarcinoma after improved evaluation of radial margin status” de Bellis et al. report a study to improve the assessment of radial margin resection status in resected perihilar cholangiocarcinoma and to investigate the effect of true R0 on survival.
This is the first assessment of its kind that prospectively assesses radial margin status with precise classification of tumor distance from the surgical margin. The authors have addressed all shortcomings that preclude an adequate and complete interpretation of these data. They provided further information on the patient characteristics of the patients examined compared to the entire cohort. The authors also indicated the number cases were an additional bile duct resection was necessary.